# SLAP: Spatial-Language Attention Policies

**Priyam Parashar**[1]  **Vidhi Jain**[2]  **Xiaohan Zhang**[1,3]  **Jay Vakil**[1]
**Sam Powers**[1,2]  **Yonatan Bisk**[1,2]  **Chris Paxton**[1]

[1]Meta  [2]Carnegie Mellon  [3]SUNY Binghamton

**Abstract:** Despite great strides in language-guided manipulation, existing work has been constrained to table-top settings. Table-tops allow for perfect and consistent camera angles, properties are that do not hold in mobile manipulation. Task plans that involve moving around the environment must be robust to egocentric views and changes in the plane and angle of grasp. A further challenge is ensuring this is all true while still being able to learn skills efficiently from limited data. We propose Spatial-Language Attention Policies (SLAP) as a solution. SLAP uses three-dimensional tokens as the input representation to train a single multi-task, language-conditioned action prediction policy. Our method shows an 80% success rate in the real world across eight tasks with a single model, and a 47.5% success rate when unseen clutter and unseen object configurations are introduced, even with only a handful of examples per task. This represents an improvement of 30% over prior work (20% given unseen distractors and configurations). We see a 4x improvement over baseline in mobile manipulation setting. In addition, we show how SLAPs robustness allows us to execute Task Plans from open-vocabulary instructions using a large language model for multi-step mobile manipulation. For videos, see the website: https://robotslap.github.io

## 1  Introduction

Transformers have demonstrated impressive results on natural language processing tasks by being able to contextualize large numbers of tokens over long sequences, and even show substantial promise for robotics in a variety of manipulation tasks [1, 2, 3]. However, when it comes to using transformers for *mobile* robots performing long-horizon tasks, we face the challenge of representing spatial information in a useful way. In other words, we need a fundamental unit of representation - an equivalent of a "word" or "token" - that can handle spatial awareness in a way that is independent of the robot's exact embodiment. We argue this is essential for enabling robots to perform manipulation tasks in diverse human environments, where they need to be able to generalize to new positions, handle changes in the visual appearance of objects and be robust to irrelevant clutter. In this work, we propose Spatial-Language Attention Policies (SLAP), that use a point-cloud based tokenization which can scale to a number of viewpoints, and has a number of advantages over prior work.

SLAP tokenizes the world into a varying-length stream of multiresolution spatial embeddings, which capture a local context based on PointNet++ [4] features. Unlike ViT-style [1], object-centric [5, 3], or static 3D grid features [2], our PointNet++-based [4] tokens capture free-form relations between observed points in space. This means that we can combine multiple camera views from a moving camera when making decisions and still process arbitrary-length sequences.

Our approach leverages a powerful skill representation we refer to as "attention-driven robot policies" [6, 7, 8, 2, 9] operating on an input-space combining language with spatial information. Unlike other methods that directly predict robot motor controls [10, 1], these techniques predict goal poses in Cartesian space and integrate them with a motion planner [6, 8, 2] or conditional low-level policy [9] to execute goal-driven motion. This approach requires less data but still has limitations, such as making assumptions about the size and position of the camera in the input scene and long training

7th Conference on Robot Learning (CoRL 2023), Atlanta, USA.

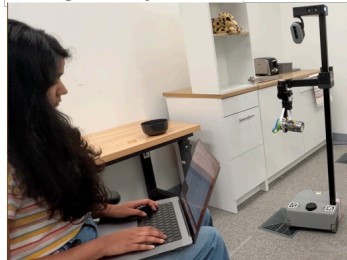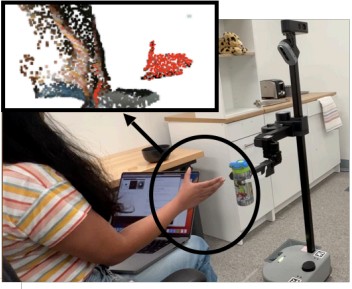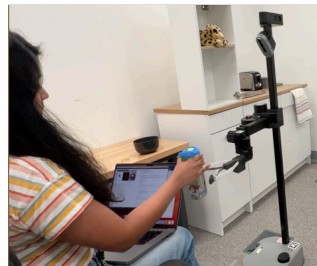
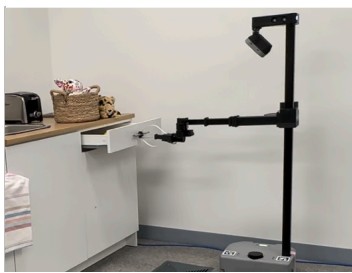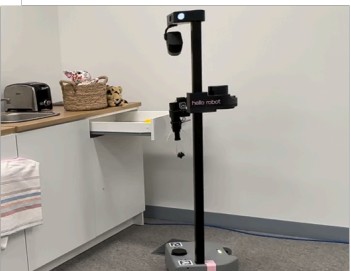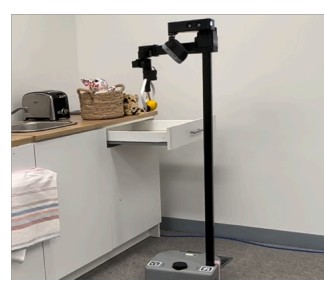

Figure 1: We propose SLAP, which allows us to learn skills for mobile manipulators to accomplish multi-step tasks given natural language goals. Our system works by training a language-conditioned *interaction prediction module*, which will determine which areas of a scene should be interacted with, in addition to an action policy which operates on predicted interaction points. This allows us to scale to more complex scenes, while predicting continuous actions.

times [7, 6, 2]. However, these methods fall into a different trap: they make strong assumptions about how big the input scene is [2], where the camera is [7, 6], and generally take a very long time to train [7, 6], meaning that they could not be used to quickly teach policies in a new environment.

SLAP uses a hybrid policy architecture. The *interaction prediction* module determines which parts of the tokenized environment the robot focuses on, and a *relative action* module predicts parameters of continuous motion with respect to the interaction features in the world. SLAP generalizes better to unseen positions and orientations, as well as distractors, while being unrestricted by workspace size and camera placement assumption, using fewer demonstrations and training in roughly a day.

We evaluate SLAP on two robot platforms. First, on a Franka Panda we can perform a direct skills comparison to the current state-of-the-art, PerAct, [2], where we demonstrate better performance with $80\%$ success rate on 8 static real-world tasks on held-out scene configurations and a $47.5\%$ success rate tested with out-of-distribution objects. Second, unlike prior work, we move beyond the stationary camera views and robot arms of a table-top setting, and demonstrate SLAP on the Hello Robot Stretch RE-2 mobile manipulator with an ego-centric camera and 6-DoF end-effector configuration. In this setting, we also include task planning to successfully execute natural language task instructions with ten demonstrations over five learned skills and three heuristic skills (Fig. 1). When SLAP is compared to the PerAct baseline for four skills in a controlled setting, we see a $4x$ improvement in the task success rate (Table 3).

## 2 Related Work

**Attention-Based Policies.** Attention-based policies have been widely studied in prior research and have been found to have superior data efficiency, generalization, and the ability to solve previously unsolvable problems [11, 9, 6, 12, 2, 13]. However, these approaches often rely on strong assumptions about the robot's workspace, such as modeling the entire workspace as a 2D image [12, 6, 7, 8] or a 3D voxel cube with predetermined scene bounds [2, 9]. This restricts their applicability and may lead to issues related to camera positioning, workspace location, and discretization size. Ad-

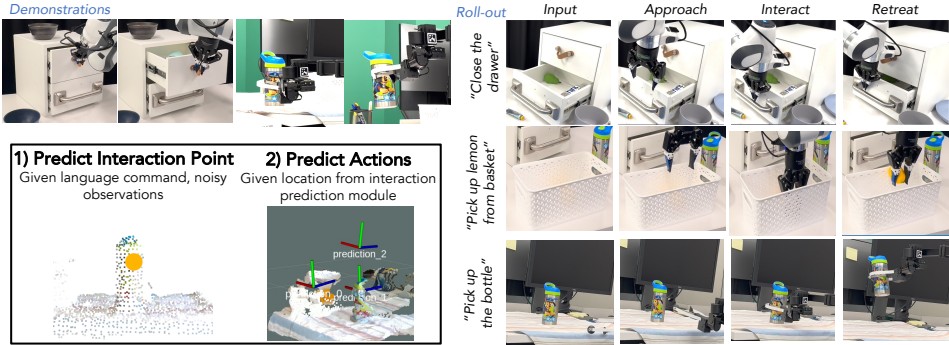

Figure 2: Spatial Language Attention Policies (SLAP) learn language-conditioned skills from few demonstrations in a wide variety of cluttered scenes. SLAP has two components: an "interaction prediction" module which localizes relevant features in a scene, and an "action prediction" module which uses local context to predict an executable action.

ditionally, these works can be seen, at least partly, as applications of object detection systems like Detic [14] or 3DETR [15], but they lack the manipulation component.

Compared to previous works, some recent studies focus on unstructured point clouds [11, 16]. These approaches demonstrate improved data efficiency and performance compared to traditional behavior cloning. For instance, Where2Act [11] and VAT-Mart [16] predict interaction trajectories, while UMPNet [17] supports closed-loop 6DoF trajectories. They share a common framework: a generalizable method to predict the interaction location and then predict local motion for the robot.

**Training Quickly with Attention-Based Policies.** CLIPort [7] and PerAct [2] are attention-based policies similar to Transporter Nets [6]. While fitting our definition of attention-based policies, they confine their workspace, use a rigid grid-like structure and treat action prediction as a discrete classification. While still a limited workspace, SPOT [12], demonstrated the usefulness of 2D attention-based policies for fast RL training, including sim-to-real transfer, and Zeng et al. [6] have shown these policies are valuable for certain real-world tabletop tasks like kitting.

**Manipulation of Unknown Objects.** Manipulation of unknown objects includes segmentation [18, 19], grasping [20, 21], placement [22], and multi-object rearrangement either from a goal image [23, 24] or from language instructions [25, 26]. These approaches rely, generally, on first segmenting relevant objects out, and then predicting how to grasp them and where to move them using separate purpose-built models, including for complex task and motion planning [27].

**Language and Robotics.** Language is a natural and powerful way to specify goals for multi-task robot systems. Several recent works [10, 1, 28] use a large-language model for task planning to combine sequential low-level skills and assume to learn the low-level skills with IL or RL. To realistically handle language task diversity, we need to learn these skills quickly. SLAP is more sample-efficient than prior IL or RL approaches. In PaLM-E [3], textual and multi-modal tokens are interleaved as inputs to the Transformer for handling language and multimodal data to generate high-level plans for robotics tasks. Our approach is a spatial extension of this strategy.

**Language for Low-Level Skills.** A number of works have shown how to learn low-level language-conditioned skills, e.g. [7, 2, 1, 29]. Like our work, Mees et al. [29] predicts 6DoF end effector goal positions end-to-end and sequences them with large language models. They predict a 2D affordance heatmap and depth from RGB; We do not predict depth, but specifically look at robustness and generalization, where theirs is trained from play data in mostly-fixed scenes. Shridhar et al. [2] predict a 3D voxelized grid and show strong real-word performance with relatively few examples, but don't look at out-of-domain generalization and are limited to a coarse voxelization of the world.

# 3  Approach

Most manipulation tasks necessarily involve interacting with environment objects [11]. We define an 'atomic skill' as a task that can be specified by an interaction point, and a sequence of relative offsets from this interaction point. For example, `pick('mug')` is an atomic skill as it can be defined in terms of an interaction point on the 'mug' and subsequent relative waypoints for approach, grasp, and lift actions. Similarly, `open('drawer')` is an atomic skill for which the interaction point is on the drawer handle, and relative waypoints from it can be defined for approach, grasp, and pull. While these examples illustrate the concept, SLAP can handle prediction of variable number of waypoints per skill.

We train a two-phase language-conditioned policy $\pi(x, l)$, which takes visual observation $x$ and a language command $l$ as inputs and predicts an *interaction point* $p_I$, as well as a set of *relative motions*, which are offsets from this point, instead of absolute coordinates. However, any realistic task given to a home robot by a user typically involves more than one atomic skill. Our system breaks down a high-level natural language task description $(\mathcal{T})$ into a sequence of atomic skill descriptions $\{l_j\}$ and uses them to condition the atomic skill motion policies. Our full paradigm is as follows:

$$\mathcal{T} \to \{l_0, ..., l_n\} \to \{\pi_j(x_j, l_j)\}_n$$
$$\forall j \in n, \quad \pi_j := (\pi_I, \pi_R), \text{ where:}$$
$$\pi_I(x_j, l_j) \to p_I \qquad \text{(3D interaction point)}$$
$$\pi_R(x_j, l_j, p_I) \to \{\mathbf{a}\}_m \quad \text{(sequence of actions)}$$

The interaction point $p_I$ is predicted by an **Interaction Prediction Module** $\pi_I$, and the continuous component of the action by a **Relative Action Module** $\pi_R$. The Interaction Prediction Module $\pi_I$ predicts *where the robot should attend to*; it is a specific location in the world, where the robot will be interacting with the object as a part of its skill, as shown in Fig. 2. $\pi_R$ predicts a relative action sequence with respect to this contact point in the Cartesian space. These actions are then provided as input to a low-level controller to execute the trajectory. These models are trained using labeled expert demonstrations; a complete overview of the training process is shown in Fig. 4. Overall, the system outputs a sequence of end-effector actions $\mathbf{a}$.

## 3.1  Scene Representation

The input observation $x$ is a structured point-cloud (PCD) in the robot's base-frame, constructed by combining the inputs from a sequence of pre-defined scanning actions. This point cloud is then preprocessed by voxelizing at a 1mm resolution to remove duplicate points from overlapping camera views. The pointcloud is then used as input into both $\pi_I$ and $\pi_R$.

For $\pi_I$, we perform a second voxelization, this time at 5mm resolution. This creates the downsampled set of points $P$, such that the interaction point $\hat{p_I} \in P$. This means $\pi_I$ has a consistently high-dimensional input and action space - for a robot looking at its environment with a set of N aggregated observations, this can be 5000-8000 input "tokens" representing the scene.

While SLAP discretizes the world similar to prior work [30, 31, 2], our approach ensures fine resolution even in larger scenes. We couple this with a set-based learning formulation which allows us to attend to fine details in a data-efficient manner.

## 3.2  Interaction Prediction Module

We use our insight about tasks being shaped around an interaction point to make learning more robust and more efficient: instead of predicting the agent's motion directly, we formulate our $\pi_I$ to solely focus on predicting a specific point $p_I \in P$, representing a single $5mm$ voxel that is referred to as the "interaction point". This formulation is akin to learning object affordance and can be thought of as similar to prior work like Transporter Nets in 2D [6]. We hypothesize that predicting attention directly on visual features, even for manipulation actions, will make SLAP

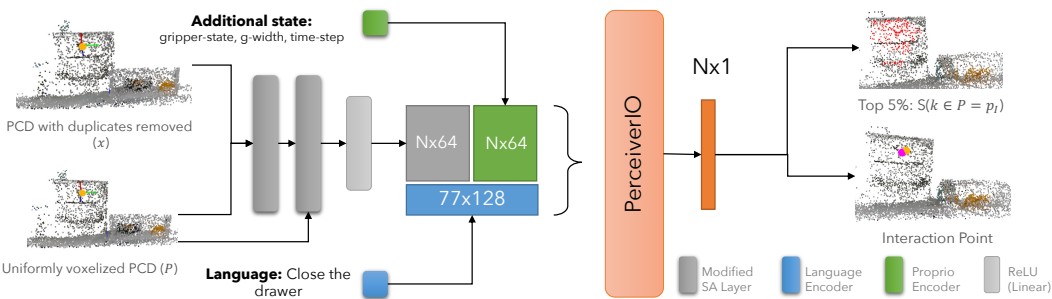

Figure 3: An overview of the architecture of the interaction prediction module. The point cloud is downsampled to remove duplicates and encoded using two modified set-abstraction layers. The SA layers generate a local spatial embedding which is concatenated with proprioceptive features - in our case, the current gripper state. Both spatial and language features are concatenated and input into a PerceiverIO transformer backbone. We then predict an interaction score per spatial feature and the *argmax* is chosen as the interaction site for command $l$.

more general. We use a PerceiverIO [32] backbone to process the data, based on prior work on language-conditioned real-world policies [2].

We first pass our input point cloud through two *modified set abstraction* layers [4] which result in a sub-sampled point-cloud with each point's feature capturing the local spatial structure around it at two different resolutions. This encourages the classifier to pay attention to *local structures* rather than a specific point that may not be visible in real-world settings. We concatenate the CLIP [33] tokenized natural language command with the encoded point cloud to create an input sequence.

Each point $i \in P$ in the point cloud is assigned a score with respect to task $\tau_j$ which results in the interaction point for that task, $p_I^j := \underset{x,y,z}{\mathrm{argmax}}(S(i = p_I^j | l, x, P, \mathcal{D}^j))$, where $\mathcal{D}^j$ is the set of expert demonstrations provided for task $\tau_j$. The IPM architecture overview is provided in Fig. 3. Note we also use binary semantic features from Detic in the Stretch experiment for training SLAP as an additional feature channel apart from the color-channels.

**Modified Set Abstraction Layer.** The default SA layer as introduced by Qi et al. [4] uses farthest point sampling (FPS) to determine which locations feature vectors are created. FPS ensures that subsampled point-cloud is a good representation of a given scene, without any guarantees about the granularity. However, it's very sensitive to the number of points selected - in most PointNet++-based policies, a fixed number of points are chosen using FPS [4]. However, SLAP must adapt to scenes of varying sizes, possibly with multiple views, and still not miss small details.

We propose an alternative PointNet++ set abstraction layer, which computes embeddings based on the original and an *evenly* downsampled version of the point-cloud, $P$. This results in a denser spatial embedding by considering a subset of all points within a certain radius of each-point in the downsampled point-cloud. This downsampled set of points guarantees we can attend to even small features, and allows us to predict an interaction point $p_I$ from the PointNet++ aggregated features.

### 3.3 Relative Action Module

The relative action module relies on the interaction point predicted by the classifier and operates on a cropped point cloud, $x_R$, around this point to predict the actions associated with this sequence. As in the interaction prediction module, the model uses a cascade of modified *set abstraction* layers as the backbone to compute a multi-resolution encoding feature over the cropped point cloud. We train three multi-head regressors (described further below) over these features to predict the actions for the overall task. Specifically, $\pi_R$ has three heads, one for each component of the relative action space: gripper activation $g$, position offset $\delta p$, and orientation $q$. Our LSTM-based architecture (details in B.2) can predict skills with variable number of actions (3,4 in our experiments).

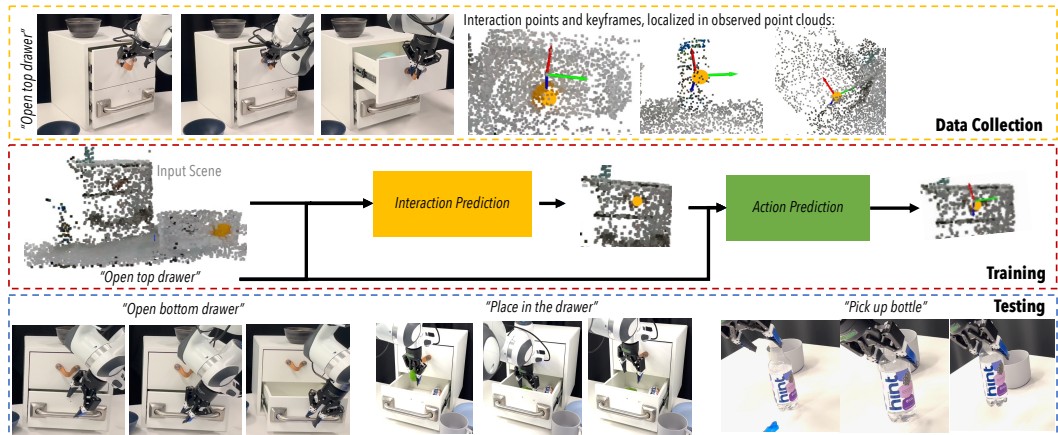

Figure 4: Illustration of the complete process for training SLAP. Demonstrations are collected and used to train the Interaction Prediction module and the Action Prediction Module separately.

Also note that the cropped input point-cloud is not perfectly centered at the ground truth interaction point $\hat{p}_I$, but rather with some noise added: $\hat{p}_I' = \hat{p}_I + \mathcal{N}(0, \sigma)$. This is done to force the action predictor to be robust to sub-optimal interaction point predictions by the interaction predictor module during real-world roll-outs. Thus, for each part of the action sequence, the keyframe position is calculated as: $p = p_I + \delta p$. When acting, the robot will move to $(p, q)$ via a motion planner, and then will send a command to the gripper to set its state to $g$.

## 3.4 Training SLAP

To collect data, an expert operator guides the robot through a trajectory, pressing a button to record *keyframes* representing crucial parts of a task. At each keyframe, we record the associated expert action $\hat{a} = (\hat{\delta p}, \hat{q}, \hat{g})$. We assume that low-level controllers exist - in our case, we use Polymetis [34] for the Franka arm and Hello Robot's controllers[1] for Stretch. Example tasks are shown in Fig. 5.

**Interaction Prediction Module.** We train $\pi_I$ with a cross-entropy loss, predicting the interaction point $p_I$ from the downsampled set of coarse voxels $P$. We additionally apply what we call a *locality loss* ($L_{loc}$), as per prior work [35]. Conceptually, we want to penalize points the further they are from the contact point, both to encourage learning relevant features as well as to aid in ignoring distractors. To achieve this, we define the locality loss as: $L_{loc} = \sum_{k \in P} \text{softmax}(f_k) \|\hat{p}_I, k\|^2$, where $f_k$ is the output of the transformer for point $k \in P$. The $softmax$ turns $f_k$ into attention over the points, meaning that $L_{loc}$ can be interpreted as a weighted average of the square distances. Points further from $\hat{p}_I$ are therefore encouraged to have lower classification scores. Combining our two losses, we obtain $L_I = CE(P, \hat{p}_I) + \frac{w}{|P|} L_{loc}$, where $w$ is a scaling constant that implicitly defines how much spread to allow in our points.

**Relative Action Module.** To train $\pi_R$, we use the weighted sum of three different losses. We train $a = (p, q, g) = \pi_R(x_R)$ with an L2 loss over the $\delta p$, a quaternion distance metric for the loss on $q$ based on prior work [36] and binary cross-entropy loss for gripper action classification (Sec. A.3).

## 3.5 Task Planning

Consider a natural language instruction from a user such as 'put away the bottle on a table'. We decompose it to a sequence of atomic skills as: `goto('bottle')`, `pick_up('bottle')`, `goto('table')`, and `place_on('table')`. We procedurally generate natural language and code templates for 16 task families. Refer A C. We use LLaMA [37] models for in-context learning [38, 39] and adapter fine-tuning [40] to learn the mapping between natural language task instructions to the corresponding sequence of atomic skills.

---

[1] https://github.com/hello-robot/stretch_ros

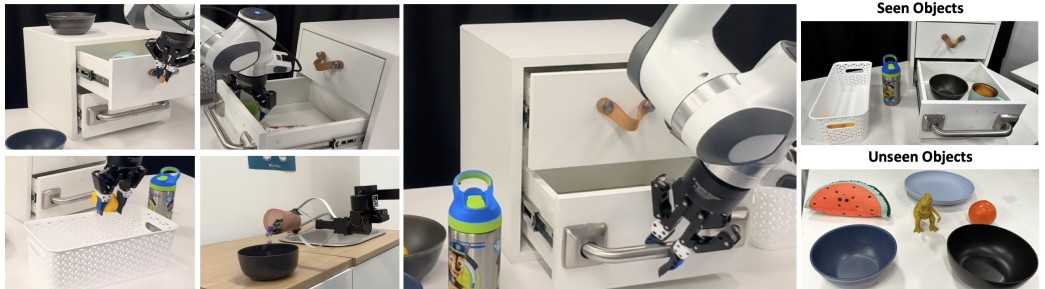

Figure 5: Examples of tasks executed on a Franka arm through our trained model in a clean setting. We trained numerous tasks (left) and tested on both seen and unseen objects (right).

# 4 Experiments

We report the success rate of our model for 8 real-world manipulation tasks in Table 2, and compare it against prior baselines trained using the same labeling scheme. Overall, we see an improvement of $1.6x$ over our best comparative baseline, PerAct [2]. Our robotics code was implemented using the HomeRobot framework [41][2]. We test each model under two different conditions: *Seen setting assumption;* i.e. those with seen distractor objects and objects placed roughly in the same range of positions and orientations as in the training data in any relative arrangement (inlcuding unseen). Second, we test under *unseen setting assumptions;* i.e. those with unseen distractor objects and the implicated object placed significantly out of the range of positions and orientations already seen. We run 5 tests per scene setting per skill per model and report the percentage success numbers in Table 2. We compare our model against Perceiver-Actor (PerAct) [2]. We train each model for the same number of training steps and choose the SLAP model based on the best validation loss. For PerAct, we use the last checkpoint, per their testing practices [2].

We also run a per-skill evaluation of SLAP and PerAct on Stretch under the *unseen setting assumption* (see Table. 3). This was accomplished by adding unseen distractor objects to the scene and moving the robot base position within reachable distance of the object. We needed to increase the workspace bounds of PerAct (1.5m cube from 1m) to capture the larger observation area for the mobile manipulator, to keep it consistent with SLAP. Note that demonstrations were collected on a different robot than the one policies were deployed on.

## 4.1 Longitudinal Task Execution on Stretch

We trained a multi-task model for the Stretch robot for five skills using 10 demonstrations each. This model was deployed in an end-to-end system which operates over code-list generated by a task-planner (as in Sec. 3.5). We ran five prompts end-to-end with four to eight skills each, using ground truth plans - we verify the viability of generating these task plans in §4.2. These experiments are done in the *unseen setting* with the robot starting anywhere with respect to the objects.

|  | GT | Inferred |
|---|---|---|
| Heuristic | 66.0 | 48.5 |
| Learned | 80.0 | 53.2 |
| Total | 68.5 | 58.5 |

Table 1: End-to-end performance. Learned skills outperform heuristics except when Detic fails.

For fair evaluation in the low-data regime, we add some structure by specifying an orthogonal viewing direction for objects. Once the agent finds the object of interest, it fires an initial prediction using SLAP to find most promising interaction point. This prediction happens under any dynamic viewing angle of the object (we assume the robot can see the object). This dynamic prediction and pre-programmed viewing angle is used to *approach* the object at an orthogonal viewing angle where the model fires an actionable prediction for the full skill execution (details see §C.3). We observe the learned skills suffering significantly when inferred plans are used to create language prompts for SLAP. This is due to discrepancies between object descriptions that the LLM generates and Detic's object detection capabilities. SLAP does not get the necessary semantic masks thus its predictions suffer. On the other hand, when semantic features

[2]https://github.com/facebookresearch/home-robot

| Skill Name | Seen | | Unseen | |
|---|---|---|---|---|
| | PerAct | SLAP | PerAct | SLAP |
| Open bottom drawer | 00% | **80%** | 00% | **60%** |
| Open top drawer | 60% | **80%** | **40%** | **40%** |
| Close drawer | **100%** | **100%** | **40%** | **40%** |
| Pick lemon from basket | 60% | **80%** | 10% | **40%** |
| Pick bottle | **60%** | **60%** | **60%** | 40% |
| Place into the drawer | 60% | **80%** | 40% | **60%** |
| Place into the basket | 40% | **100%** | 10% | **60%** |
| Place into the bowl | 40% | **60%** | 00% | **40%** |
| Average Success Rate | 50% | 80% | 27.5% | 47.5% |
| Improvement | | 1.6x | | 1.7x |

Table 2: SLAP and PerAct [2] performance on real world Franka manipulation tasks. We evaluate both seen scenes (seen object positions and distractors), but in different arrangements, and unseen scenes with previously-unseen object positions and distractors. SLAP is notably better overall in both conditions.

| Skill Name | PerAct | SLAP |
|---|---|---|
| Open Drawer | 0% | 60% |
| Close Drawer | 40% | 100% |
| Take bottle | 0% | 80% |
| Pour into bowl | 40% | 80% |

Table 3: SLAP on a mobile manipulator using a multi-task model across 4 skills, over 5 tries. With semantic predictions added to our feature space, we see the model perform better on unseen scenes with new distractors and unseen relative position of the robot with respect to the scene

from Detic are available IPM performance significantly improves even with unseen distractors (80% against 47.5% in Table 2). We still see failures when relative position is significantly perturbed. Please see §E.1 for ablation analysis of our design against PerAct.

## 4.2 Task planning with in-context learning and fine-tuning LLaMa

Previous work has shown the strength of language models as zero-shot planners [42], a result strengthened by improved techniques for "in-context learning" or prompting [43]. To verify that models can produce task plans with the skills we defined, we experiment with both in-context learning (IC) [44] of LLaMA [37] and adapter fine-tuning (FT) [40]. Table 4 presents the mod-

| LLaMa | | Verb | Noun | Task Acc. | Corr. | Lat. (sec.) |
|---|---|---|---|---|---|---|
| IC | 7B | 83 | 81 | 76 | 61 | 16.4 |
| IC | 30B | 81 | 81 | 76 | 62 | 27.6 |
| FT | 7B | 100 | 98 | 99 | 91 | 19.5 |

Table 4: Fine-tuning (FT) outperforms in-context learning (IC) for same latency.

els' verb, noun, and combined accuracies. Task Correctness is a binary score if the entire prediction was correct, and finally, latency is measured in seconds on a single A6000 with 16 GB RAM.[3] High Task Correctness from a small model verifies the compatibility of our skills with LLM task planning.

## 5   Limitations

SLAP has high variance in out-of-distribution situations, resulting in complete failure if $\pi_I$ fails to correctly identify the context. For $\pi_R$, multimodal or noisy data still poses issues; replacing $\pi_R$ with a policy which can better handle this data, e.g. Diffusion Policies [46]. Overall system has multiple points-of-failure due to heuristic policies, unaligned language and vision models; end-to-end trainable architectures and cross-modal alignment could help.

## 6   Conclusion

We propose a new method for learning visual-language policies for decision making in complex environments. SLAP is a novel architecture which combines the *structure* of a point-cloud based input with *semantics* from language and accompanying demonstrations to predict continuous end-effector actions for manipulation tasks. We demonstrate SLAP on two hardware platforms, including an end-to-end evaluation on a mobile manipulator, something not present in prior work. SLAP also outperforms previous state-of-the-art, PerAct, on both mounted and mobile robot setups.

---

[3]Adaptor fine-tuning increases the model size by ∼6%, which accounts for the additional latency compared to IC. We use standard inference libraries so results are comparable, but not optimized for runtime [45].

## 7 Acknowledgements

We thank Sriram Yenamandra for his help working with the Stretch codebase and running real-robot navigation and grasping. More generally, we thank the HomeRobot team for software support [41].

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

# Appendix

## Table of Contents

## A  Training

Below is expanded information on our training, algorithm, and data processing to improve reproducibility.

### A.1  Data Collection and Annotation

When collecting an episode with the Franka arm, we first scan the scene with a pre-defined list of scanning positions to collect an aggregated $x$. In our case, we make no assumption as to what or how many these are, or how large the resulting input point cloud $x$ is. With the Hello Robot Stretch [47], we collect data based on exactly where the robot is looking.

Then, we collect demonstration data using kinesthetic teaching for the Franka arm (demonstrator physically moves the robot) and via controller teleoperation for the Stretch robot. The demonstrator moves the arm through the trajectory associated with each task, explicitly recording the *keyframes* [48] associated with action execution. These represent the salient moments within a task – the bottlenecks in the tasks' state space, which can be connected by our low-level controller.

## A.2 Data Processing

We execute each individual skill open-loop based on an initial observation. We use data augmentation to make sure even with relatively few examples, we still see good generalization performance.

**Data Augmentation.** Prior work in RGB-D perception for robotic manipulation (e.g. [18, 49]) has extensively used a variety of data augmentation tricks to improve real-world performance. In this work, we use three different data augmentation techniques to randomize the input scene $x$ used to train $p_I = \pi_I(x, l)$:

- *Elliptical dropout:* Random ellipses are dropped out from the depth channel to emulate occlusions and random noise, as per prior work [50, 18]. Number of ellipses are sampled from a Poisson distribution with mean of 10.

- *Multiplicative Noise:* Again as per prior work [50, 18, 22], we add multiplicative noise from a gamma process to the depth channel.

- *Additive Noise:* Gaussian process noise is added to the points in the point-cloud. Parameters for the Gaussian distribution are sampled uniformly from given ranges. This is to emulate the natural frame-to-frame point-cloud noise that occurs in the real-world.

- *Rotational Randomization:* Similar to prior work [2, 22, 25], we rotate our entire scene around the z-axis within a range of $\pm 45$ degrees to help force the model to learn rotational invariance.

- *Random cropping:* with $p = 0.75$, we randomly crop to a radius around $\hat{p}_I + \delta$, where $\delta$ is a random translation sampled from a Gaussian distribution. The radius to crop is randomly sampled in $(1, 2)$ meters.

**Data Augmentation for $\pi_R$.** We crop the relational input $x_R \subset x$ around the ground-truth $p_I$, using a fixed radius $r = 0.1m$. We implement an additional augmentation for learning our action model. Since $p_I$ is chosen from the discretized set of downsampled points $P$, we might in principle be limited to this granularity of response. Instead, we randomly shift both $p_I$ and the positional action $\delta p$ by some uniformly-sampled offset $\delta r \in \mathbb{R}^3$, with up to $0.025m$ of noise. This lets $\pi_R$ adapt to interaction prediction errors of up to several centimeters.

## A.3 Action Prediction Losses

Following [36] for the orientation, we can compute the angle between two quaternions $\theta$ as:

$$\theta = \cos^{-1}(2\langle \hat{q}_1, \hat{q}_2 \rangle^2). \tag{1}$$

We can remove the cosine component and use it as a squared distance metric between 0 and 1. We then compute the position and orientation loss as:

$$L_R = \lambda_p \| \delta p - \hat{\delta p} \|_2^2 + \lambda_q (1 - \langle \hat{q}, q \rangle) \tag{2}$$

where $\lambda_p$ and $\lambda_q$ are weights on the positional and orientation components of the loss, set to 1 and $1e - 2$ respectively.

Predicting gripper action is a classification problem trained with a cross-entropy loss. For input we use the task's language description embedding and proprioceptive information about the robot as input, i.e. $s = (l, g_{act}, g_w, ts)$ where $g_{act}$ is 1 if gripper is closed and 0 otherwise, $g_w$ is the distance between fingers of the gripper and $ts$ is the time-step. The gripper action loss is then:

$$L_g = \lambda_g CE(g, \hat{g}) \tag{3}$$

where $\lambda_g$ is the weight on cross-entropy loss set to 0.0001. The batch-size is set at 1 for this implementation.

We train $\pi_I$ and $\pi_R$ separately for $n = 85$ epochs. At each epoch, we compare validation performance to the current best - if validation did not improve, we reset performance to the last best model.

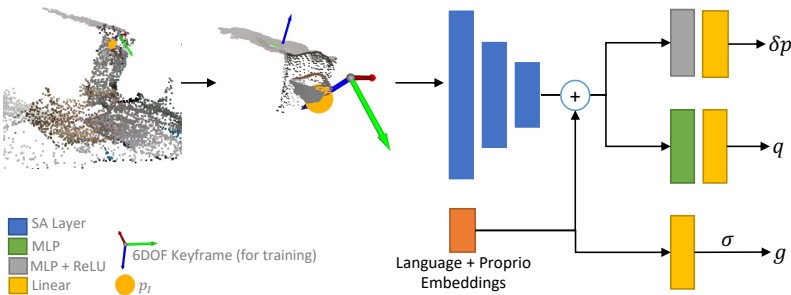

Figure 6: Regression model architecture with separate heads for each output. The point-cloud is cropped around the interaction point with some perturbation and passed to a cascade of set abstraction layers. Encoded spatial features are then concatenated with language and proprioception embeddings to predict position offset of action from interaction point, absolute orientation and gripper action as a boolean.

### A.4 Skill Weighting

In Stretch experiments, we used a wide range of skills with different error tolerances and corresponding variances. As a result, we needed to use two different sets of weights for learning position, orientation and gripper targets. A more forgiving weight-set for noisier tasks like pouring and handover, and a tighter weight-set for task with more consistent trajectories like opening and closing the drawer. These weights were empirically determined but can be further optimized via hyperparameter tuning methods.

### A.5 Training for PerAct and SLAP

In all our experiments we ensure PerAct and SLAP are trained on the same *data volume*. Data volume is defined as total number of augmented samples per collected sample. Note this results in different number of training steps per model. This is due to the LSTM-based model updating once per trajectory while PerAct updates once per sample in a trajectory.

## B    Relative Action Module

In our work, the Relative Action Module $\pi_R$ is assumed to be some *local* policy which predicts end-effector poses. In our case, we implement two different versions of this policy, one which was used on the static Franka manipulator and one which was implemented on the Stretch. In both cases:

- The policy predicts an *end effector pose* relative to the predicted interaction point from $\pi_I$
- The policy is conditioned on a *local crop* around this interaction point.

### B.1    MLP Implementation

Fig. 6 gives an overview of the MLP version of the regression model. The model takes in the cropped point cloud (augmented during training as discussed in Sec. A.2. We saw that injecting random noise to the interaction point during training allowed the policy to, at test time, recover from failures (because it predicted an interaction point *near* the correct area, instead of at the correct position).

### B.2    LSTM-based Implementation

We observed the MLP architecture suffer when positional distribution of actions varied widely with respect to the interaction point position across tasks due to the multi-modality this introduced. Thus we modified the above model by adding an LSTM to condition each task and action with a hidden-

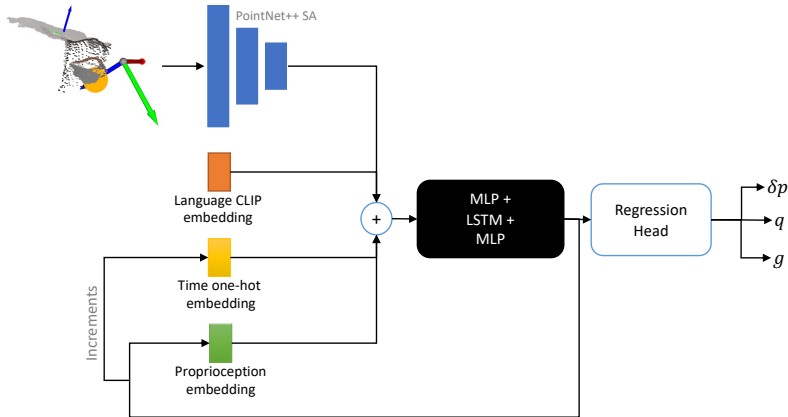

Figure 7: LSTM-based regression model architecture based on the regression head and PointNet++ embeddings introduced in Fig. 6. LSTM-based architecture shows higher stability in learning action distribution with wider distribution due to the conditioning effect.

state. This model exhibited better performance in learning wider action distribution, based on our initial experiments with the outlined Stretch tasks.

## C  High-level Task Planning and Execution

### C.1  Dataset

We procedurally generated a dataset consisting of more than 500k tuples of language instructions and the corresponding sequence of atomic skills. The data is created for 16 task families and can be extended further in the future. For each task family, 10% samples are held-out for evaluation. The distribution of samples across task families is shown in 5.

For each task family, we define a corresponding template containing the sequence of atomic skills. This means that the sequence of atomic skill "verbs" is the same among the samples of a task family. Each sample within a task family differs in terms of language instructions and object(s) of interaction.

To populate these templates and generate the data, we create a list of more than 150 movable objects kitchen objects, surfaces like `table`, `kitchen counter` and articulated objects like `drawer`, `cabinets`. For `pour` skill, we create a list of "spillable" items such as `cup of coffee`, or `bowl of jelly beans`. Similarly, for wipe skill, we have a list of items to wipe with such as `sponge`, or `brush`.

### C.2  Models

Table 4 shows that in-context learning with 5 examples from the same task family achieves close to 76% accuracy. There is no training involved with in-context learning, so it can't overfit. For finetuning, our evaluation consists of a held-out dataset with unseen variations in the phrasing of the language instruction and/or object(s) of interaction. The goal of the fine-tuned model is to demonstrate that it is possible to achieve improved task planning performance with lower latency than in-context learning with larger models. We do not evaluate the generalization of the fine-tuned model with unseen task families in this work. The remaining results in this work use the fine-tuned model for task planning.

Table 5: Data Distribution of procedurally generated samples across task families.

| Task Family | Total number of samples |
|---|---|
| Bring X From Y Surface To Pour In Z Then Place On W Surface | 35640 |
| Bring X From Y Articulated To Wipe Z | 612 |
| Move X From Y Surface To Z Surface | 16092 |
| Move X From Y Surface To Z Articulated | 301400 |
| Take X From Human To Z Articulated | 19300 |
| Bring X From Y Surface To Human | 5933 |
| Bring X From Y Surface To Wipe Z | 3168 |
| Take X From Human To Pour In Z | 2184 |
| Take X From Human To Z Surface | 8050 |
| Take X From Human To Wipe Z | 1458 |
| Move X From Y Articulated To Z Articulated | 97923 |
| Bring X From Y Articulated To Human | 13617 |
| Bring X From Y Surface To Pour In Z | 3960 |
| Bring X From Y Articulated To Pour In Z | 8712 |
| Move X From Y Articulated To Z Surface | 50060 |
| Take X From Human To Pour In Z And Place On Y Surface | 19656 |
| | **587765** |

## C.3 System Architecture and Plan Executionr

We use SLAP with three heuristic policies: SEARCHFOROBJECT ($\pi_{search}$), PICKUPOBJECT ($\pi_{pick}$) and PLACEON ($\pi_{place}$). $\pi_{search}$ uses Detic, frontier-based exploration policy and a language query to explore the map until object described by the language query is found in the view of the robot. This is also one of the primary points of failure when LLM is integrated into the pipeline. LLM expects Detic to be able to handle any freeform query of the object (for example, "detect open drawer" is a typical output from the LLM however always fails as Detic has no notion of an open drawer).

$\pi_{pick}$ is a heuristic picking policy which always grabs given object from the top given its mask from Detic and depth from camera. This is a generally robust policy but fails when *contextual, task-oriented grasps* are required in a task. For example consider a bottle placed within a cabinet, the only pick policy which will solve this scene is one where gripper grasps bottle laterally and not from the top. $\pi_{place}$ places whatever object is in robot's gripper on the surface previously detected by Detic. Another point of failure, since up-close Detic is not able to detect objects like table, counter surfaces, etc.

SLAP is used *after* the object is successfully detected by Detic, given language query, and is in the robot's field of view. Note the robot at this point can be at any unconstrained orientation and position with respect to the object, the only condition sufficed here is that the object is within sight. We use SLAP's interaction prediction module to estimate the affordance over this object with Detic's mask as an additional feature, and predict an interaction point on the object. The robot then uses hand-engineered standoff orientations to move to a head-on position with respect to the object, where we use the full SLAP system to predict the action trajectory.

The standoff orientations are used so that SLAP can be tested fairly within reasonable bounds of the state distribution it was trained for. Training data for all our skills is recorded from a very narrow range of robot positioning with respect to the objects (see Appendix D.1). This means the action prediction module's sense of action orientation is not robust to huge rotational variations over object's position around the robot's egocentric frame. Interaction prediction module on the other hand is very robust as it does not need to consider directionality, just the local structure of object and related affordance. See Fig. 8 for details on how regions are assigned to specific objects, and related stand-off orientations provided to the robot so that it is always facing the object head-on after estimating the interaction point. Given predicted interaction point $p_i$, pre-determined standoff

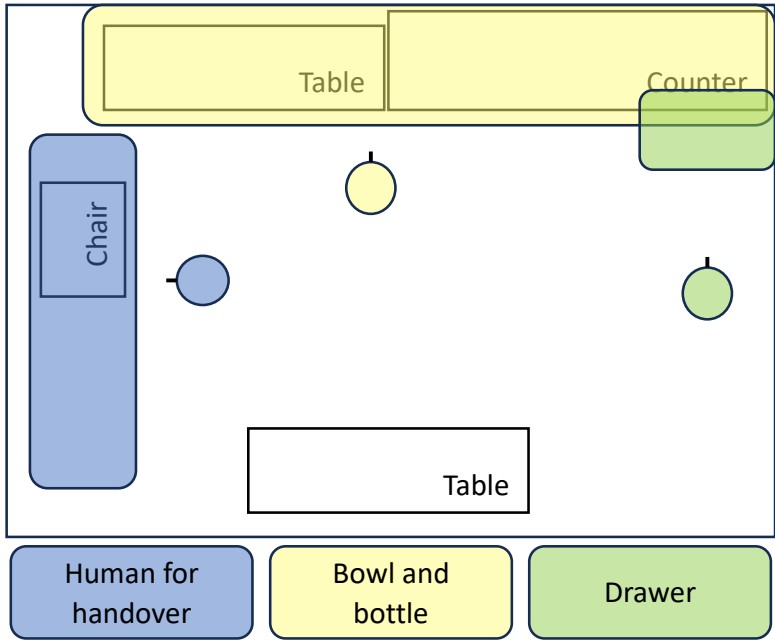

Figure 8: Diagram showing where immovable objects were placed in the environment (note that chair can move anywhere within the blue region). The three colored regions signify the placement assignment for different artifacts involved in the tasks. The circular symbol signifies the robot's pre-determined orientation where the beak represents where the robot will be facing. Position for robot's placement is determined based on predicted $p_i$

orientation vector $vec_{standoff}$ and pre-determined 3D standoff distance vector $dist_{standoff}$, the robot moves to an orientation $vec_{standoff}$ and position:

$$pos_{next} = p_i + dist_{standoff} * (-1 * vec_{standoff})$$

The orientation vector is of the form $(1, 0, 0)$ or $(0, 1, 0)$ in this evaluation.

## D  Experimental Setup for Skills

Here we refer to atomic skills learned by SLAP as simple tasks or "tasks". This allows us to discuss corresponding "actions" that are defined in terms of the relative offset from the interaction point.

### D.1  In vs. Out Of Distribution

We used a number of objects for our manipulation experiments, which included both in- and out-of-distribution objects (see Fig. 9 and Fig. 10). One goal of SLAP is to show that our methods generalize much better than others to different types of scenes and different levels of clutter.

We also randomized the objects with seen and unseen clutter around them, as well as placed them in unseen environments. Fig. 12 and Fig. 13 show the extent of variation captured in test scenes against training scenes for Stretch experiments. Fig. 11 shows a range of test scenes from Franka evaluations.

### D.2  Skill Definitions and Success Conditions

Every real-world task scene had a sub-sample of all within-distribution objects. Following describes skills and their success condition for Franka experiments:

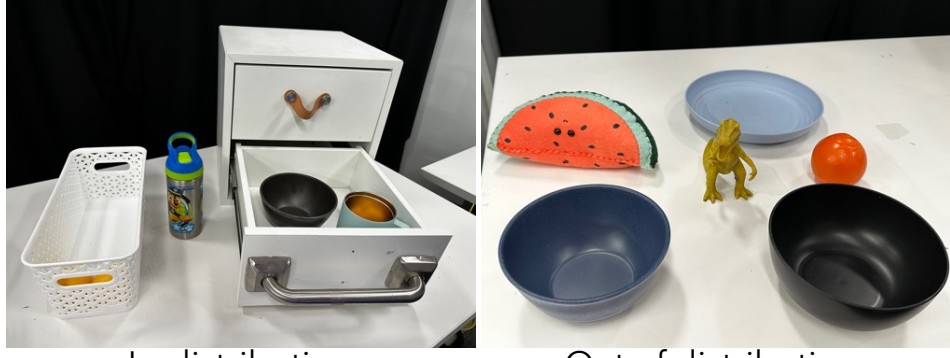

In-distribution                    Out of distribution

Figure 9: Within distribution objects used at training time and out-of-distribution objects introduced during testing in our experiments.

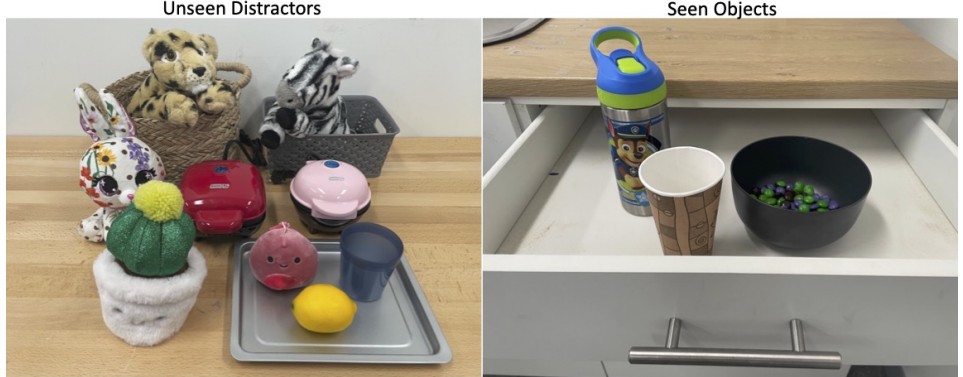

Figure 10: Seen objects and unseen distractors used in longitudinal experiments with Stretch.

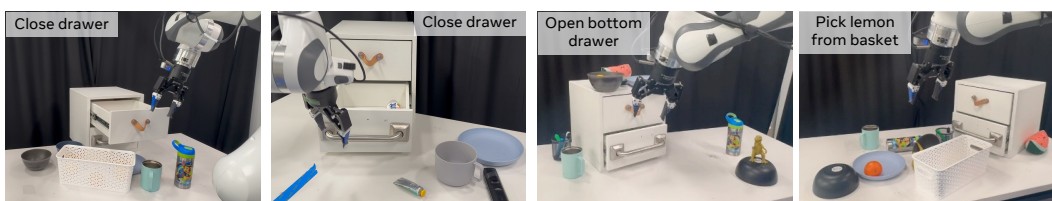

Figure 11: Snapshot of test scenes from Franka evaluations to show the range of variation at test-time

1. Open the top drawer
   - *Task:* Grab the small loop and pull the drawer open. Drawer configuration within training data is face-first with slight orientation changes
   - *Action labeling:* Approach the loop, grab the loop, pull the drawer out
   - *Success metric:* When the drawer is open by 50% or more

2. Open the bottom drawer
   - *Task:* Grab the cylindrical handle and pull the drawer open. Drawer configuration within training data is face-first with slight orientation changes. Note significantly different grasp is required than for top drawer
   - *Action labeling:* Approach the handle, grab it, pull the drawer out
   - *Success metric:* When the drawer is open by 50% or more

3. Close the drawer

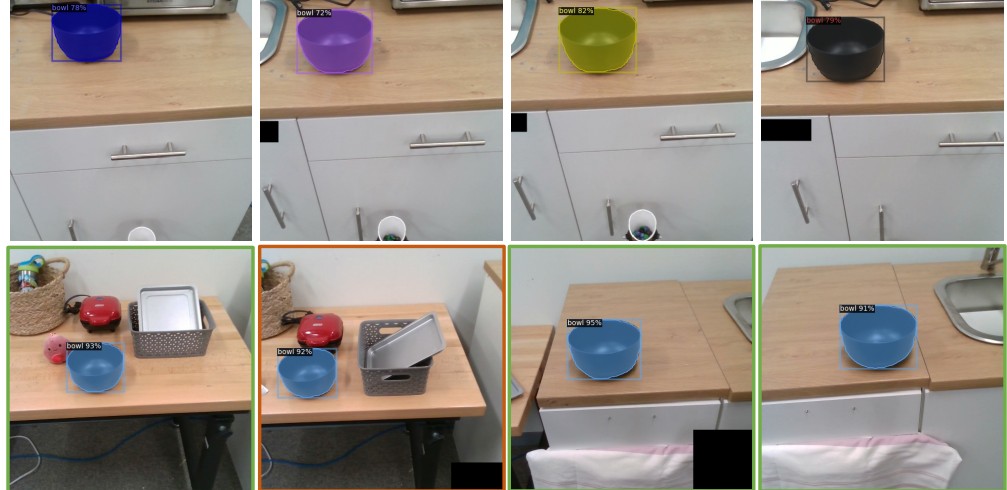

Figure 12: Pour into bowl task: Showing variability and out-of-domain distribution covered by test against training samples. Top row: Training scenes. Note that the bowl was always placed somewhere on this particular region (right of sink) on the counter. Bottom row: Test scenes. Note bowl is surrounded by unseen clutter, placed in novel unseen environment and at relatively different positioning with respect to the camera and robot. Green boundary signifies successful episode, red a failed episode.

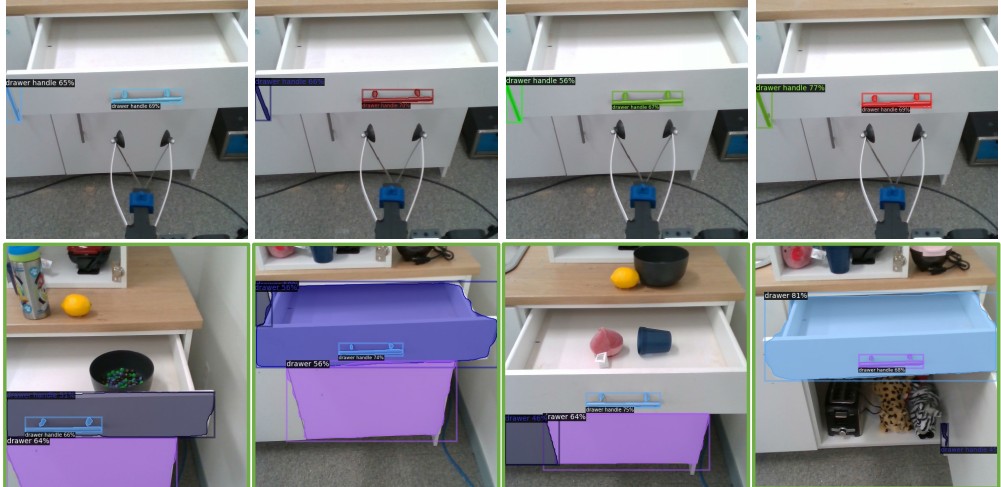

Figure 13: Close drawer task: Showing variability and out-of-domain distribution covered by test against training samples. Top row: Training scenes. Note the absence of any clutter and narrow range of relative positioning of drawer with respect to the camera and robot. Bottom row: Test scenes. Note presence of objects used in other tests in the same frame. Green boundary signifies successful episode, red a failed episode.

- *Task:* This task is unqualified, i.e. the instructor does not say whether to close the top or bottom drawer instead the agent must determine which drawer needs closing from its state and close it. Align the gripper with the front of whichever drawer is open and push it closed. The training set always has only one of the drawers open, in a front-facing configuration with small orientation changes
- *Action labeling:* Approach drawer from the front, make contact, push until closed
- *Success metric:* When the drawer is closed to within 10% of its limit or when arm is maximally stretched out to its limit (when the drawer is kept far back)

4. Place inside the drawer

- *Task:* Approach an empty spot inside the drawer and place whatever is in hand inside it
- *Action labeling:* Top-down approach pose on top of the drawer, move to make contact with the surface and release the object, move up for retreat
- *Success metric:* Object should be inside the drawer

5. Pick lemon from the basket

- *Task:* Reach into the basket where lemon is placed and pick up the lemon
- *Action labeling:*
- *Success metric:* Lemon should be in robot's gripper
- *Considerations:* Since the roll-out is open-loop and a lemon is spherical in nature, a trial was assigned success if the lemon rolled out of hand upon contact after the 2nd action. This was done consistently for both PerAct and SLAP.

6. Place in the bowl

- *Task:* Place whatever is in robot's hand into the bowl receptacle
- *Action labeling:* Approach action on top of the bowl, interaction action inside the bowl with gripper open, retreat action on top of the bowl
- *Success metric:* The object in hand should be inside the bowl now

7. Place in the basket

- *Task:* Place the object in robot's hand into the basket
- *Action labeling:* Approach action on top of the free space in basket, interaction action inside the basket with gripper open, retreat action on top of the basket
- *Success metric:* The object is inside the basket

8. Pick up the bottle

- *Task:* Pick up the bottle from the table
- *Action labeling:* Approach pose in front of the robot with open gripper, grasp pose with gripper enclosing the bottle and gripper closed, retreat action at some height from previous action with grippers closed
- *Success metric:* The bottle should be in robot's gripper off the table

Notably, success for opening drawers is if the drawer is 50% open after execution; this is because sometimes the drawer is too close to the robot's base for it to open fully with a fixed-base Franka arm.

Following describes the skills undertaken in Stretch experiments and their success conditions:

1. Open the drawer

- *Task:* Grab the handle of the drawer and pull the drawer open
- *Action labeling:* Approach the handle, grab the handle, pull the drawer out and open grasp
- *Success metric:* When the drawer is open by 100%

2. Close the drawer

- *Task:* Align with the surface of drawer's face and push the drawer close
- *Success metric:* When the drawer is closed within 10% of fully-closed configuration

3. Pour into bowl

- *Task:* Skill starts with a cup filled with candies already in robot's gripper. Align cup with the bowl and turn the cup in a pouring motion
- *Success metric:* When $\geq 50\%$ of candies are in the bowl

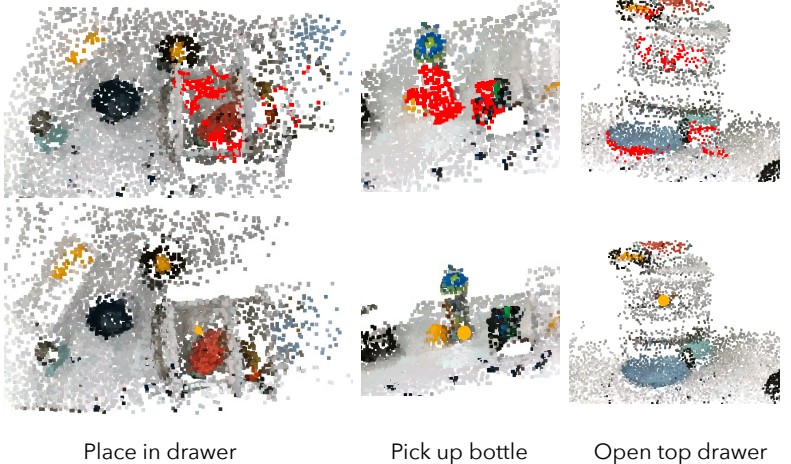

| Place in drawer | Pick up bottle | Open top drawer |

Figure 14: Examples of out of distribution predictions made by $\pi_I$. We show that it is able to handle heavy clutter around the implicated object to predict interaction points. Note that the prediction for bottle picking is sub-optimal in this example.

4. Take bottle
   - *Task:* Approach bottle with gripper orientation in the right configuration, grasp the bottle, lift the bottle up and retract keeping the grasp
   - *Success metric:* The bottle is in robot's gripper in a stable configuration at the end of execution
5. Handover to person
   - *Task:* Approach hand of the person with object in hand, align with hand's surface and release the object, finally retracting the gripper back
   - *Success metric:* The object is in human's hand at the end of execution

Note that we count the success for pouring $\geq 50\%$ of the candies because we are comparing this task to pouring a liquid. Liquid would pour out completely in the intended final configuration due to different dynamics.

### D.3 Language Annotations

In the following, we include the list of language annotations used in our experiments. Table 6 shows the language that was used to train the model; we're able to show some robustness to different language expressions. We performed a set of experiments on held-out, out-of-distribution language despite this not being the focus of our work; this test language is shown in Table 7.

### D.4 Out of distribution Results from SLAP

We show more results for the attention point predicted by $\pi_I$ in Fig. 14. For the placement task, the agent has never seen a heavily cluttered drawer inside before, but it is able to find flat space that indicates placement affordance. For the bottle picking task, this sample has a lemon right next to the bottle which changes the shape of the point-cloud around the bottle. We see that $\pi_I$ is able to find an interaction point albeit with placement different from expert and lower down on the bottle. Similarly, the open-top drawer sample has more heavy clutter on and around the drawer to test robustness.

Fig. 15 shows the prediction and generated trajectory for picking up a previously unseen bottle. Note that while the models are able to detect the out of distribution bottle, the trajectory actually fails because the bottle is much wider and requires more accuracy in grasping. For the mobile manipulator domain, we observe SLAP performed better than vanilla PerAct on every count. Our

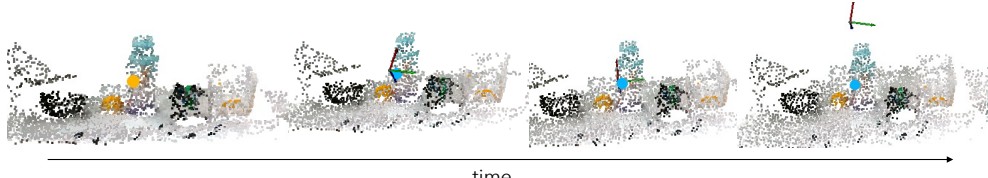

time

Figure 15: A generalization example of success for our model. The new bottle has same shape as the within distribution bottle but is much taller, different in color and wider in girth. The model is able to predict the interaction site and a feasible trajectory around it. We note though the execution of this trajectory was a failure; due to wider girth of the bottle the predicted grasp was not accurate enough to enclose the object.

hypothesis is that SLAP's better performance is due to the addition of semantic features, more efficient training, and higher resolution due to a non-grid point-cloud representation.

### D.5 Motion Planning Failures

Our evaluation system has a simple motion planner which is not collision aware as a result we saw a number of task failures for both the models. However, we note that the frequency of task failures due to motion planning problems was higher for PerAct. We think it is because PerAct predicts each action of the same task as an entirely separate prediction trial, while SLAP forces continuity on the relative motions for the same task by centering them around the interaction point (see Fig. 15). That said, we also note with a collision-aware motion planner PerAct may not run into such issues as seen during our evaluations. However the planner setting and conditions were same across both models in these evaluations. The authors note in their own paper their heavy reliance on good motion planning solutions [2].

## E  Additional Analysis

### E.1  Ablations

**Hybrid vs Monolithic Architecture (Table-top).** We train SLAP and PerAct such that they observe same amount of data. SLAP outperforms PerAct on six of eight tasks when tested in in-distribution settings and five of eight tasks in out-of-distribution settings on Franka. PerAct performs equally well as our model for two of eight tasks on our in-distribution scenes. Similarly, for our "hard" generalization scenes, PerAct performed equally well in two cases, and actually outperformed SLAP when picking up a bottle. Under similar experimentation conditions, SLAP outperforms PerAct in all four tasks in cluttered scenes for the mobile manipulator environment on Stretch. In failure cases, $\pi_R$ predicted the correct trajectory, but not with respect to the right part of the object.

**Unseen Scene Generalization.** We see a drop in the success rate for both PerAct and SLAP when tested on out-of-distribution settings. PerAct would often predict the correct approach actions, but then it would fail to grasp accurately. With SLAP, however, we saw that $p_I$ was predicted fairly accurately, but the regressor would fail for out-of-distribution object placements specifically because of bad orientation prediction. When $\pi_I$ failed, it was because the position and orientation of the target object was dramatically different, *and* unseen distractors confused it. We see better results for SLAP under Stretch setting due to the addition of semantic features from Detic.

### E.2  Visualizing the Learned Attention

Since we use scores to choose the final interaction point, our classifier model is naturally interpretable, being able to highlight points of interest in a scene. We visualize this attention by selecting the points with the highest 5% of interaction score given a language command $l$.

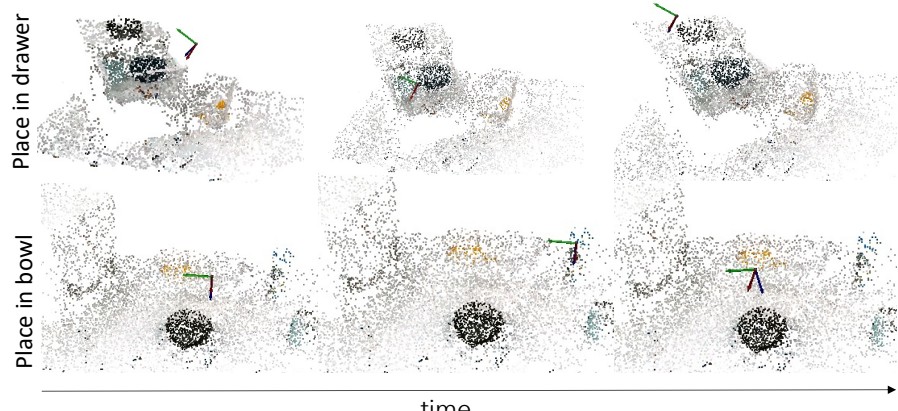

Figure 16: Examples of failure cases for our baseline, PerAct, for the "place in drawer" and "place in bowl" tasks. In the top example, the gripper is moved from drawer's side towards inside, instead of from the top as demonstrated by expert. The gripper ends up pushing off the drawer to the side as our motion-planner is not collision-aware. Note that SLAP does not exhibit such behaviors as $\pi_R$ implicitly learns the collision constraints present in demonstrated data. In the bottom example, each action prediction is disjointed from previous and semantically wrong.

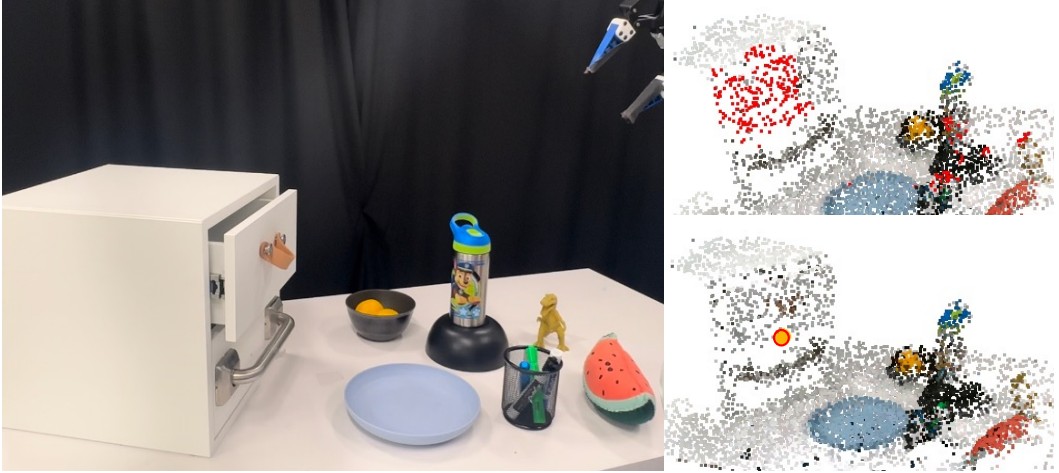

Figure 17: An example out-of-distribution SLAP failure where an extreme sideways configuration of the drawer is paired with unseen distractors for the "open top drawer" skill. Note that the attention mask ranks other distractors in its top 5% and fails to choose an optimal interaction point.

### E.3 Language Generalization

By using pretrained CLIP language embeddings to learn our spatial attention module $\pi_I$, our model can generalize to unseen language to some extent. We tested this by running an experiment where we evaluate performance on in-distribution scene settings, prompted by a held-out list of language expressions. We choose three representative tasks for this experiment and run 10 tests with 2 different language phrasings.

## F  Additional Related Work

We note some other related work related to the larger language-conditioned, mobile manipulation domain that SLAP is situated in, but not as directly relevant.

**Vision-Language Navigation.** Similar representations are often used to predict subgoals for exploration in vision-language navigation [30, 31, 8, 51, 52]. HLSM builds a voxel map [30], whereas FiLM builds a 2D representation and learns to predict where to go next [31]. VLMaps proposes an object-centric solution, creating a set of candidate objects to move to [8], while CLIP-Fields learns an implicit representation which can be used to make predictions about point attentions in responds to language queries [51], but does not look at manipulation. Similarly, USA-Net [52] generates a 3D representation with a lot of semantic features including affordances like collision. Such a representation can naturally be incorporated for collision-aware action plans at prediction time.

| Task Name | Training Annotations |
|---|---|
| pick up the bottle | pick up a bottle from the table |
| | pick up a bottle |
| | grab my water bottle |
| pick up a lemon | pick the lemon from inside the white basket |
| | grab a lemon from the basket on the table |
| | hand me a lemon from that white basket |
| place lemon in bowl | place the lemon from your gripper into the bowl |
| | add the lemon to a bowl on the table |
| | put the lemon in the bowl |
| place in the basket | place the object in your hand into the basket |
| | put the object into the white basket |
| | place the thing into the basket on the table |
| open bottom drawer | open the bottom drawer of the shelf on the table |
| | pull the second drawer out |
| | open the lowest drawer |
| close the drawer | close the drawers |
| | push in the drawer |
| | close the drawer with your gripper |
| open top drawer | open the top drawer of the shelf on the table |
| | pull the first drawer out |
| | open the highest drawer |
| place in the drawer | put it into the drawer |
| | place the object into the open drawer |
| | add the object to the drawer |

Table 6: Examples of language used to train the model.

| Task Name | Held-Out Test Annotations |
|---|---|
| Pick up the bottle | Grab the bottle from the table |
| | Pick up the water bottle |
| Open the top drawer | Pull top drawer out |
| | Open the first drawer |
| Place into the drawer | Add to the drawer |
| | Put inside the drawer |

Table 7: Examples of out-of-distribution language annotations used for evaluation

