# OpenReview forum: "SLAP: Spatial-Language Attention Policies"
_robot-learning.org/CoRL/2023/Conference — CoRL 2023 Poster_

### Official Review · Reviewer_91vF · 2023-07-15

**Confidence:** 4
**Originality:** Good
**Technical Quality:** Good
**Clarity Of Presentation:** Good
**Impact:** 3

**Recommendation:**

Weak Accept: I recommend accepting the paper, but will not argue for my recommendation if the majority of other reviewers have a different opinion.

**Review:**

Overall, the paper presents an interesting and well-engineered system for manipulation tasks. The overall method design makes sense but with some small minor limitations on the experiment design.

1.	In Table 2, the number of evaluations per task is only 5, which may cause large statistical variations. One success and failure could lead to 20% of the difference. Considering the performance difference for SLAP and PerAct is only 30%, having only 5 trials per tasks is clearly not significant for a thorough comparison.

2.	PerAct is designed to be a multi-task policy. I am not sure comparing PerAct in a single task learning is fair?

3.	More details on atomic task design. Since the training data is composed by the experts, I am a bit confused about how to generate that many 500k samples in a reasonable time and labour budget? How many raw language instructions were collected and how many atomic skills per task is generated?



**Quality Of The Limitations Section:**

Limitations are addressed clearly

**Questions For Rebuttal:**

1. More experiments. Maybe add additional baselines like pure BC, multi-task learning setting; more evaluations to improve statistical significance.
2. More explanations of the atomic skill design.

**Robotics Focus:**

Sufficient demonstration on hardware

**Summary Of Paper:**

This paper presents a language-conditioned policy for mobile and table-top manipulation tasks. The proposed policy is based on spatial attention on point clouds and can be decomposed into two stages: i) an interaction point is predicted based on sub-sampled point cloud, and ii) relative action is predicted around the predicted interaction point. The proposed method is shown to perform well in real-world manipulation tasks compared to a competitive baseline PerAct.



**Summary Of Recommendation:**

Good method design with a bit insufficient experiments. Could be improved with rebuttal.

---

> ### Author Response · Authors · 2023-08-14
> **Follow-up to rebuttal**
>
> We thank the reviewer for their time and effort. We wanted to follow-up and check if we were successful in addressing your concerns, and if there might be any other details that we can provide.

---

### Official Review · Reviewer_zJZk · 2023-07-17

**Confidence:** 4
**Originality:** Very Good
**Technical Quality:** Very Good
**Clarity Of Presentation:** Good
**Impact:** 3

**Recommendation:**

Weak Accept: I recommend accepting the paper, but will not argue for my recommendation if the majority of other reviewers have a different opinion.

**Review:**

This paper presents a pretty exciting and novel robot learning system that performs well in around 10 real-world robotics tasks and is able to handle unseen distractor objects and novel object configurations to some degree during test time.

This work has a few noteworthy strengths. First of all, the algorithmic design seems novel and interesting. Although spatial action map / attention-based policies have been studied before, this paper is one of its first kind to directly operate on 3D point cloud with modified set abstraction layers adapted from PointNet++ and also take advantage of transformers’ ability to take in varying-length stream of multi-resolution spatial embeddings.

Secondly, by decomposing the policy into two stages (“finding interaction point”, and “generating relative motion around that point”), the SLAP demonstrates better performance and sample efficiency than previous methods on real world robotics tasks. The experimental evaluation (per skill and multiple skill setting) is relatively thorough. The proposed model is able to generalize to unseen objects and object poses, which is quite impressive.

Thirdly, the paper is relatively well-written. Different components of the system are well explained (the two major modules, how LLM is used, how the point clouds are processed, etc). The figures are pretty illustrative, as well as the videos in the supplementary material.

This work, however, is not without limitations. Overall, although the paper is relatively easy to follow, it seems to be missing some important information (more details can be found in the section “Questions For Rebuttal” below). For example, the paper should probably include more results analysis, especially the failure mode analysis (e.g. which of the two modules fail more often, in seen and unseen settings, and why? What percentage of the failures are due to motion planning failures? Why is the motion planner not collision aware). The paper also misses citations to other attention-based policies, such as ReLMoGen[1] and Spatial Action Maps for Mobile Manipulation[2]. ReLMoGen is particularly relevant because it also predicts a 3D interaction point (a pixel mapped to 3D space from depth sensing) and a local nonprehensile interaction from that point (predicting desired offset), although it’s trained with RL rather than IL. Lastly, the paper should probably tone down its claim about “mobile manipulation”. One of the primary challenges of mobile manipulation, compared to tabletop manipulation, is that 1) the scene is only partially observable (i.e. the desired interaction point might be outside the current FoV), 2) whole body motion (e.g. the robot might need to move the arm and the base holistically when opening a fridge door in a narrow kitchen). This work tackles neither of these two cases, and it’s left unexplained how the robot navigates to objects (those in FoV and outside FoV).

Nonetheless, this work seems to be fairly promising. With proper revision, this work can be very impactful for the skill learning and the overall robot learning community

[1] Xia, Fei, et al. "Relmogen: Integrating motion generation in reinforcement learning for mobile manipulation." 2021 IEEE International Conference on Robotics and Automation (ICRA). IEEE, 2021.
[2] Wu, Jimmy, et al. "Spatial action maps for mobile manipulation." arXiv preprint arXiv:2004.09141 (2020).


**Quality Of The Limitations Section:**

Limitations are addressed clearly

**Questions For Rebuttal:**

- Add missing citations to related work such as ReLMoGen and Spatial Action Maps for Mobile Manipulation as mentioned above.
- Add more error analysis to share more insights about your experiments
- In Fig 3, what is the top 5% prediction used for? Just for visualization?
- How are the semantic features from Detic being used exactly? It’s not very well-explained in the paper. Why is it only used in the Stretch experiments, but not the table top experiments? Can we have an ablation study for models with and without semantic features?
- What would happen if the predicted action (i.e. delta pose) is out of reach for the robot or leads to collision with the environment? Would the episode immediately terminate and labeled as a failure? Is LSTM-based Relative Aciton Module closed-loop, and how? Does it recover from its mistakes?
- How many demonstrations are collected on Franka and Stretch robot respectively?
- It’s mentioned that the two modules are trained separately. But isn’t the second module (relative action module) depending on the first module (interaction prediction module) for cropping the input point cloud? More training procedure details are appreciated.
- What exactly are the hand-specified orthogonal viewing directions for in the Stretch robot? Why is it needed? What if we take them out?
- How is navigation handled for objects in FoV and outside FoV?
- Since the relative action module is aggregating global features from SA layers and outputting a single desired 6 dof delta pose, it’s a bit doubtful how fine-grained/dextrous the action can be. Is it really at the “mm” level accuracy already? Have the authors thought of some sort of voting mechanism for neighboring points to collectively vote for an action? See VoteNet [3]

[3] Qi, Charles R., et al. "Deep hough voting for 3d object detection in point clouds." proceedings of the IEEE/CVF International Conference on Computer Vision. 2019.



**Robotics Focus:**

Sufficient demonstration on hardware

**Summary Of Paper:**

The paper proposes Spatial-Language Attention Policies (SLAP), which leverages three-dimensional tokens (features extracted from point clouds) as input presentation to train a single multi-task, language conditioned action prediction policy. The method is comprised of two primary modules: Interaction Prediction Module, which predicts an interaction point given a language instruction and input point clouds, and Relative Aciton Module, which predicts a desired end-effector pose relative to the interaction point and the desired gripper state given the interaction point, cropped point cloud and language instruction. The two modules are trained separately with imitation learning. The whole system demonstrates promising results in table-top pick-and-place tasks (including opening/closing articulated objects) and mobile manipulation pick-and-place tasks, and outperformed previous baselines.

**Summary Of Recommendation:**

Based on the review above, I recommend Weak Accept. I am open to changing my mind based on the rebuttal.

---

> ### Comment · Reviewer_zJZk · 2023-08-13
> **Thank you for your response.**
>
> Dear authors,
>
> I would like to thank you for your thorough response to my questions/comments. I think they address most of my concerns and questions and are very helpful and informative.

---

> > ### Author Response · Authors · 2023-08-14
> > **Follow-up**
> >
> > We are happy to know our answers and explanations were helpful and informative for you, and we thank you for your time and effort in reviewing the additional information. We wanted to follow-up and check if we have addressed all your concerns, and if there might be any other details that we can provide.

---

### Official Review · Reviewer_iD6X · 2023-07-19

**Confidence:** 4
**Originality:** Good
**Technical Quality:** Good
**Clarity Of Presentation:** Fair
**Impact:** 3

**Recommendation:**

Weak Accept: I recommend accepting the paper, but will not argue for my recommendation if the majority of other reviewers have a different opinion.

**Review:**

Strengths:
1. The presentation of the method is clear and easy to follow.
2. Good coverage of related work.
3. The design of the two-phase language-conditioned policy to predict the affordance and the waypoints is a viable option.
4. Using demonstrations from a different robot than the robot used for deployment suggests good generalization abilities.
5. Real-world experiments on two robots, Franka and Stretch.
6. Honest discussion of the limitations.

Weaknesses:
1. In lines 92-94, the authors criticize PerAct for not looking at out-of-domain generalization, but the current evaluation section does not look at it.'
2. It would be interesting to compare the method to 2D attention-based policies, e.g., SPOT.
3. The method constructs an input point cloud from pre-defined scanning actions (lines 117-118) followed by voxelization. How much time is needed to build this scene representation? Is it required for each atomic skill? How much does this affect method deployment?
4. What are the benefits of the proposed set abstraction layer?
5. The evaluation is reported for 5 tests per scene setting per skill per model. The number of scenes is not reported, and it seems from Table 2 that there were only 5 runs (aside from the 10% for PerAct).
6. It would be interesting to show the results with the last checkpoint of SLAP. It will be a fair comparison, and it will demonstrate robustness. While using the model with the highest validation loss is equivalent to cherry-picking.
7. The robot is spawned randomly in the environment (line 222). Are the objects randomized during the experiments?
8. Section 4.2. how big is the dataset? What are the evaluation settings? Table (Figure 5) suggests potential overfitting.
9. The ablation study shows that SLAP can not generalize well for out-of-distribution settings which contradicts the claims in the introductions.

**Quality Of The Limitations Section:**

Limitations are addressed clearly

**Questions For Rebuttal:**

Please see the weaknesses section above.

**Robotics Focus:**

Sufficient demonstration on hardware

**Summary Of Paper:**

The paper presents Spatial-Language Attention Policies (SLAP), a method for language-guided mobile manipulation. SLAP has two components (1) an interaction prediction module that localizes interaction points in a point cloud (object affordance), and (2) a relative action module that predicts relative action sequence w.r.t. the contact points in the cartesian space. The models are trained with labeled expert demonstrations.
The approach uses atomic skills to manipulate objects, defined by the interaction point and subsequent relative waypoints. A language-conditioned policy predicts the atomic skills parametrization given input observations.
The authors evaluated the method with 5 test runs per scene setting per skill per model, comparing it to Perceiver-Actor (PerAct) method and demonstrating superior performance.

**Summary Of Recommendation:**

The paper proposes an interesting method for language-conditioned policies operating on point clouds.
The current experimental sections need some revisions to support the paper's claims.
We vote for rejecting the current version of the paper, but we will support it for acceptance if the authors can address our comments.

**Update after rebuttal:**

The authors addressed most of our comments reasonably; we raised our recommendation to a weak accept.

---

### Official Review · Reviewer_1D6s · 2023-07-20

**Confidence:** 4
**Originality:** Good
**Technical Quality:** Fair
**Clarity Of Presentation:** Very Good
**Impact:** 3

**Recommendation:**

Weak Reject: I recommend rejecting the paper, but will not argue for my recommendation if the majority of other reviewers have a different opinion.

**Review:**

Strengths:

- The proposed hierarchical design is reasonable. Empirically, it demonstrates to outperform the original PerAct in 11/16 testing tasks (if counting each instruction with seen and unseen objects to be two tasks).

- The proposed method is clearly motivated and described in the paper. The technical part of the paper is easy to follow.

Weaknesses:

- One of my biggest concerns is about the experiment setup. In the experiments, each of the tasks/instructions is separately designed in a different environment. This does not convincingly demonstrate if the learned policy can control the robot to solve tasks faithful to the comannded instruction. Based on our previous experience working on language-conditioned imitation learning, it is usually relatively easy to find certain scenes and a single instruction in each of those scenes, which allows the learned policy to solve the task. But if the policy is commanded by other different instructions for another plausible task in the same scene, the policy can fall short. For example, when there is an open drawer in front of the robot, closing the drawer can be the obvious thing for the robot to do given the training data. But if the robot is commanded to "Move the mug into the drawer" at the same state, the robot might either performs some random irrelevant behaviors or still close the drawer regardless. We believe that a more principled way to evaluate such language-conditioned policies would be to design N environments and, in each environment design M tasks (e.g. N = 3, M = 3) and shows if the robot can successfully follow different instructions at exactly the same initial state. For example, in the "Close drawer" tasks shown on the website, other tasks such as "Move the bottle to the drawer" and "Move the stuffed animal to the drawer" need to be also evaluated. Otherwise, it seems not entirely convincing if the policy successfully gains the ability to follow different instructions.

- Some of the details about the method and the experimental setup seem to be missing. For example, how many training trajectories were collected? I couldn't find it either in the main paper or in Appendix A.1. And What are the hyperparameters and how are they chosen?

- The results show that the proposed method does not generalize very well to unseen objects. The success rates seem to be much lower than those with seen objects.

- It would be better if ablation on the design choices such as the modified set abstraction layer and the combination of the two losses for the interaction prediction module could be included in the experiments. Otherwise, it would be hard to justify the significance of these design choices.

**Quality Of The Limitations Section:**

Limitations are addressed clearly

**Questions For Rebuttal:**

- How many training trajectories were collected?

- What are the hyperparameters and how are they chosen?

**Robotics Focus:**

Sufficient demonstration on hardware

**Summary Of Paper:**

This paper proposes a method based on PerAct to train robots to follow natural language instructions given observed point clouds. One of the main contributions is a hierarchical architecture design which first predicts a reference point using a reference layer and then predicts the actions accordingly. The proposed method is compared with the original PerAct on a set of table-top manipulation tasks and achieves better performances in 11 of the 16 designed tasks.

**Summary Of Recommendation:**

I like the proposed method and some of the demos look nice. However, the current experiments do not seem to be sufficient to justify the claims and contributions in the paper. Therefore, I am leaning towards rejection.

---

> ### Author Response · Authors · 2023-08-14
> **Follow-up to rebuttal**
>
> We thank the reviewer for their time and effort. We wanted to follow-up and check if we were successful in addressing your concerns, and if there might be any other details that we can provide.

---

> ### Comment · Reviewer_1D6s · 2023-08-15
> **Response to Rebuttal**
>
> I appreciate the authors' rebuttal. However, the answer to my concern regarding _"Each of the tasks/instructions is separately designed in a different environment. This does not convincingly demonstrate if the learned policy can control the robot to solve tasks faithful to the comannded instruction."_ is completely orthogonal to what this question originally asks about. The question is about how the experimental results could convincingly justify if the trained policy can solve different tasks when commanded by **different instructions** in **the same environment**, while your answer explains why the performance of the trained policy is worse in **unseen environments** given **the same instruction**.
>
> I was wondering if the authors could provide any further explanation and experimental results to showcase that the trained policy can follow different instructions in the same scene (starting at the same initial condition)? Without such experimental results, it would be hard to justify whether the policy is truly language-conditioned.

---

> > ### Author Response · Authors · 2023-08-16
> > **Author Response**
> >
> > Authors appreciate that large majority of our answers made sense to the reviewer.
> >
> > Apologies for not understanding your question, we were making a case about how the unseen scenes have all the task-implicated objects (~6) within them and we still observe the policy executing the task ~47.5% of the time. Although not the exact same, all of these scenes have *all the objects* SLAP was trained on and system is tested on different language descriptions.
> >
> > The seat of our language-conditioned affordance learning is with the interaction prediction module. There could be an experiment where we  train just the interaction prediction module and ask it to predict different affordances on the same scene as a working response towards one part of this question. Does this seem more aligned to the question you were asking? Due to the limited time we express regret that such an experiment, although possible, is not feasible within the rebuttal window.
> >
> > Another observation from the past: when the interaction prediction module predicted a suboptimal interaction point it was usually on the surface of the drawer. We observed that under such conditions the action prediction module (non-LSTM) predicting actions relevant to “opening the drawer” instead as the cropped input was more aligned to opening data rather than closing. There can be an extension of the above-mentioned experiment to test action prediction module’s sensitivity to language and structure.

---

> > > ### Comment · Reviewer_1D6s · 2023-08-16
> > > **Response to rebuttal**
> > >
> > > Thank you for your responses. However, the case you were making does not answer directly the question asked above. Unless the authors could provide evidence showing that the policy can follow **different instructions** in **the same environment** (starting at the same state), the main claims in the paper are not convincingly justified.

---

### Comment · Area_Chair_SNZr · 2023-08-11
**Discussion until Aug 15th**

I would like to thank the authors and reviewers and would like to encourage you to make best use of the discussion period which will end *Aug 15 at 11:59 PM PT*.

In particular, to the reviewers: you are highly encouraged to engage in additional discussions with the authors if there are any remaining questions you would like the authors to elaborate on further.

---

### Author Response · Authors · 2023-08-11
**General Comment**

We thank the reviewers for their feedback. We are glad that the reviewers liked our method (R1), finding it “novel and interesting” (R3, R4) and the execution well-engineered (R4). R3 specifically notes how our architecture design is “one of its first kind to directly operate on 3D point cloud” and exhibits “better sample efficiency than previous methods on real world robotic tasks”. Reviewers generally appreciated our method being deployed on real-world robots noting that we demonstrate our method on two different embodiments (R2) outperforming PerAct 11/16 (R1), the fact that demonstrations were collected on a different robot than the one used for deployment (R2) and R3 calling them “thorough” and the generalization “impressive”. Majority of the reviewers agree that our paper was well-written and easy to follow with “clearly motivated” (R1) and “clear presentation” (R2) of method, “good coverage of related work” (R2), the “different components [being] well explained” (R3) and “honest discussion of limitations” (R2). Reviewers also appreciate our demo videos (R1, R3) and the illustrative visuals in the paper (R3).

The reviewers highlight that there is some miscommunication in our explanation of experimental design and how this creates a disconnect between our experiments and our claims. The reviewers raise several good points in their questions and note aspects which are actually strengths of our paper but we failed to communicate clearly. We have quickly made several changes to share additional results and videos but will take longer to make all the rhetorical and explanatory changes for the final camera ready.

Note on experiments: We want to note that we conducted further comparative experiments between SLAP and PerAct by deploying PerAct on Stretch under the same experimental conditions as submitted experiments. We found that on a mobile manipulator where the input point-cloud is unconstrained, SLAP performs 4x better than PerAct. PerAct solves none of the episodes for 2 tasks (open drawer, take bottle), scoring 40% successes for closing the drawer and pouring from glass into bowl. Note we needed to change the grid-size of PerAct from 1mx1mx1m to 1.5mx1.5mx1.5m for a reasonable voxel-grid representation of the inputs.

Our revised paper includes the results outlined above and a number of new visuals to answer queries from the reviewers. We have also updated the paper to address the information gaps highlighted by the reviewers. We are working on including related work pointed out in the reviews.

---

### Decision · Program_Chairs · 2023-08-30

**Decision:**

Accept (Poster)

**Comment:**

The authors propose an approach to allow robots for follow natural language instructions that relies on a multi-resolution tokenization of point-cloud-based features. The review process resulted in a recommendation of "weak accept" by a majority of 3 reviewers and "weak reject" by 1 reviewer.

The reviewers mentioned the following strengths of the work in particular:
- improved performance vs PerACT was demonstrated
- the overall approach appears to be well-founded and motivated
- the paper is well written and easy to follow
- the decomposition of the policy into two phases was appreciated as novel

Among the opportunities for further improvement, the reviewers remarked on
- potential difficulty of the approach to generalize
- an evaluation showcasing that different instructions can be followed in the same scene and starting configuration was requested
- a comparison to 2D attention-based policies may be valuable
- additional simulation evaluations (as in PerACT0 may have been of value

Given the overall positive reviews, and since the work has promise in a rapidly evolving area of robotics, I believe the paper is above the acceptance threshold and of interest to the CoRL community despite potential areas that could be improved further.